# Morphological and Hemodynamic Changes during Cerebral Aneurysm Growth

**DOI:** 10.3390/brainsci11040520

**Published:** 2021-04-19

**Authors:** Emily R. Nordahl, Susheil Uthamaraj, Kendall D. Dennis, Alena Sejkorová, Aleš Hejčl, Jaroslav Hron, Helena Švihlová, Kent D. Carlson, Yildirim Bora Suzen, Dan Dragomir-Daescu

**Affiliations:** 1Department of Mechanical Engineering, NDSU, Fargo, ND 58108, USA; emily.nordahl@ndsu.edu (E.R.N.); bora.suzen@ndsu.edu (Y.B.S.); 2Division of Engineering, Mayo Clinic, Rochester, MN 55905, USA; Uthamaraj.Susheil@mayo.edu (S.U.); Dennis.Kendall@mayo.edu (K.D.D.); 3Department of Neurosurgery, Masaryk Hospital, J. E. Purkyně University, 401 13 Usti, Czech Republic; alena.sejkorova@gmail.com (A.S.); ales.hejcl@gmail.com (A.H.); 4International Clinical Research Center, St. Anne’s University Hospital, 656 91 Brno, Czech Republic; 5Institute of Experimental Medicine AS CR, 142 20 Prague, Czech Republic; 6Faculty of Mathematics and Physics, Mathematical Institute, Charles University, Ke Karlovu 3, 121 16 Prague, Czech Republic; jaroslav.hron@mff.cuni.cz (J.H.); helena.svihlova@seznam.cz (H.Š.); 7Department of Physiology and Biomedical Engineering, Mayo Clinic, Rochester, MN 55905, USA; Carlson.Kent@mayo.edu

**Keywords:** hemodynamics, aneurysm growth, computational fluid dynamics, oscillatory shear index, wall shear stress, kinetic energy

## Abstract

Computational fluid dynamics (CFD) has grown as a tool to help understand the hemodynamic properties related to the rupture of cerebral aneurysms. Few of these studies deal specifically with aneurysm growth and most only use a single time instance within the aneurysm growth history. The present retrospective study investigated four patient-specific aneurysms, once at initial diagnosis and then at follow-up, to analyze hemodynamic and morphological changes. Aneurysm geometries were segmented via the medical image processing software Mimics. The geometries were meshed and a computational fluid dynamics (CFD) analysis was performed using ANSYS. Results showed that major geometry bulk growth occurred in areas of low wall shear stress (WSS). Wall shape remodeling near neck impingement regions occurred in areas with large gradients of WSS and oscillatory shear index. This study found that growth occurred in areas where low WSS was accompanied by high velocity gradients between the aneurysm wall and large swirling flow structures. A new finding was that all cases showed an increase in kinetic energy from the first time point to the second, and this change in kinetic energy seems correlated to the change in aneurysm volume.

## 1. Introduction

With the widespread utilization of non-invasive axial imaging, unruptured aneurysms are a common finding. Identification of risk factors for rupture is a key to select patients for treatment. Aneurysm growth at follow-up is a well-known risk factor for aneurysm rupture [1], yet there is a paucity of studies focused on hemodynamic factors predisposing to growth, and limited information on hemodynamic changes associated with aneurysm growth over time. Recently, there has been a focus on developing computational fluid dynamic (CFD) models to investigate the hemodynamics of cerebral aneurysms. There are currently two major contrasting viewpoints as to whether a low or high flow influences the growth of aneurysms [1,2]. To highlight the current uncertainty in aneurysm rupture prediction, an international simulation challenge was held recently [3]. For this challenge, 17 separate research groups were asked to perform CFD simulations of five aneurysms found in one patient, and to predict which one of the aneurysms ultimately ruptured. The only patient information provided to participants were the 3D rotational angiography images of the aneurysms. Groups were encouraged to use whatever hemodynamic and/or morphological parameters they deemed important to predict the rupture. Only 4 of the 17 groups correctly predicted which aneurysm ruptured. This challenge demonstrated (1) that the research community lacks consensus on which parameters are important for rupture risk prediction, and (2) that there is a need for a broader, deeper knowledge base regarding the correlation between such parameters and aneurysms.

To fully understand aneurysm rupture, it is important to understand the growth of aneurysms and how this growth is related to hemodynamic parameters. Only a few existing studies have dealt specifically with aneurysm growth, and there is limited information available on hemodynamic changes associated with aneurysm growth over time. The goal of the present study was to investigate changes in the hemodynamic parameters of four cerebral aneurysms that demonstrated interval growth during follow-up, and to see how these parameters changed as the aneurysms grew.

## 2. Materials and Methods

Four unique patient aneurysms were chosen for study based on their location and size, to investigate hemodynamic changes with geometrical differences caused by aneurysm growth. For this retrospective study, Digital Imaging and Communications in Medicine (DICOM) Magnetic Resonance Angiography (MRA) images were obtained from patient databases at two time points for each patient: the first time point was at the initial diagnosis and the second was at follow-up. The sizes (diameters) and years of growth for each aneurysm are represented in Table 1.

Using the medical image processing software Mimics (Materialise, Leuven, Belgium), we implemented a segmentation strategy that retained artery inlet and outlet lengths upstream and downstream of the aneurysm to ensure model accuracy [4]. ANSYS ICEM CFD meshing software was used to create hexahedral element meshes for each model, and CFD simulations were performed using ANSYS Fluent (ANSYS, Canonsburg, Pennsylvania, USA). A mesh sensitivity study was conducted for each geometry using three different mesh element sizes and comparing the CFD solutions obtained with each mesh size to determine when the solution had converged [5]. The converged mesh was then used for further analysis. Blood was assumed to be a homogeneous Newtonian fluid. A Womersley flow profile was prescribed at the inlet to model pulsatile flow. The velocity versus time profile at the inlet center, normalized by the value at peak systole, is shown in Figure 1 for one cardiac cycle. The peak systole value in each model was adjusted until the models achieved the desired average flow rate. We specified an average inlet flow rate of 240 mL/min for models of ICA aneurysms (P1 and P2), and an average flow rate of 150 mL/min for models of MCA aneurysms (P3 and P4) [6]. These representative values were used because patient-specific blood flow data were not available. Traction-free boundary conditions were approximated at all outlets by setting each outlet to a relative pressure of zero. The arterial walls were considered rigid and no-slip. For all computations, the cardiac cycle was assumed to have a period of one second, and a solution time step size of 0.0001 s was used. Four cardiac cycles were simulated for each model to minimize transient numerical errors. The results shown in the next section are from the fourth cycle; instantaneous results (velocity and WSS) are shown at peak systole, and the oscillatory shear index (OSI) was computed over the entire cycle.

Specific data planes were set up for visual comparison of each case that bisected the impingement jet at the aneurysm neck. Several hemodynamic parameters were calculated from each simulation [7,8]: wall shear stress (WSS), oscillatory shear index (OSI), and total kinetic energy (KE). WSS and OSI were calculated at all points on the aneurysm bleb surface, and changes in these quantities from one time point to the next were evaluated using averages over the bleb surface. KE was calculated at every point within the volume of the bleb, and changes in KE were computed using the sums of the values within the bleb at each time point. All hemodynamic and morphological changes given below were normalized by dividing the respective changes by the aneurysm growth time in years (last column of Table 1). This was done to convert changes to “average change per year” values that enable normalized comparisons of these changes over all the aneurysms, which had growth times ranging from one year to over seven years. To compare aneurysm growth and remodeling, morphological changes in an aneurysm bleb were observed as a directional, areal, or volumetric growth. Shape remodeling was observed as a change in aneurysm topology that can occur with or without aneurysm growth at a specific location.

## 3. Results

The changes in the hemodynamic parameters considered in this study are summarized in Table 2. To provide a general sense of the growth of these aneurysms, the changes in common morphological parameters are also included in this table. The morphological parameter abbreviations used are size ratio (SR), aspect ratio (AR), ellipticity index (EI) and non-sphericity index (NSI).

### 3.1. P1 Aneurysm

Overall, P1 grew in volume 61% per year, with large growth in its width (Figure 2). Substantial remodeling occurred near the impingement region, which was located in the artery below the aneurysm neck. A pointed growth occurred opposite of impingement area, increasing the ellipticity (Figure 2a). Significant aneurysm growth also occurred on the side opposite of impingement.

Figure 3 shows the pathlines as well as the contours of velocity on the impingement data plane, as well as the WSS and OSI on vessel and aneurysm walls for the initial and follow-up models of P1. The velocity and WSS results are shown at peak systole of the fourth cardiac cycle, and the OSI results were computed using the fourth cardiac cycle results. The pathlines suggest a larger swirling flow structure near the aneurysm wall with another smaller swirling region through the center, with little change at follow-up (Figure 3a,b). However, KE increased by 256% per year (Table 2). Velocity gradients on the side of the bleb opposite the impingement region also increased at follow-up (Figure 3a,b). Large growth occurred in areas of high velocity gradient between the wall and the larger swirling flow structure, indicated by the red arrow in Figure 3b.

Areas of large remodeling in the impingement region showed significant decreases in WSS (black arrow in Figure 3d). However, on the bleb surface away from the neck, the average WSS increased slightly between the initial and final time points. The black circle in Figure 3d shows an area where the WSS gradients increased from the initial to the final time point. An area of low WSS is seen near the center of flow rotation (Figure 3c). The maximum OSI at follow-up was similar to the initial value, but the size of the elevated OSI region decreased at follow-up, resulting in a change of OSI of −33% per year (Figure 3e,f).

### 3.2. P2 Aneurysm

Overall, P2 grew in volume an average of 56% per year, with large growth in height. The width of the aneurysm grew opposite of the impingement zone (Figure 4). Two new blebs formed, one on top and another near the neck. In addition, neck remodeling led to a third small bleb. Similar to P1, the pathlines suggest a large swirling flow structure near the aneurysm wall and another more concentrated swirling region developing at the center (Figure 5a,b). For P2, however, the increase in KE was smaller than P1, with a change of 56% per year (see Table 2). The largest bleb growth occurred in an area of low velocity next to the main swirling flow structure (red arrow in Figure 5b).

Figure 5d,f show changes in WSS contours correlated to areas of large remodeling in the impingement region (black arrow in Figure 5d). Increases in areas of high WSS with overall large gradients were found near the impingement region. Regions of significant aneurysm growth correlated to areas of low WSS at both time points (black circle in Figure 5d). Elevated OSI was found near the impingement area and next to the daughter bleb formation (Figure 5e,f). Areas of elevated OSI appear to increase from the initial to final time points, but the average increase in OSI was only 9% per year.

### 3.3. P3 Aneurysm

P3 grew in volume 11% per year (Figure 6). Small daughter bleb formations were found near and opposite from the impingement region.

Figure 7a,b indicate that the pathlines changed more over time for P3 compared with P1 and P2, and generally suggest a larger swirling flow structure near the elongated aneurysm wall in addition to two smaller regions of rotation. The increase in KE for P3 was small (7% per year). Large growth occurred in areas of high velocity gradient between the wall and the larger region of rotation, similar to P1. These areas were found above and across from the impingement region (red arrow in Figure 7b).

There is an increase in the area of elevated WSS near the impingement region, where a daughter bleb also forms (Figure 7c,d). The black arrow in Figure 7d indicates both daughter bleb formation and the elevated WSS region. Large growth occurred in areas of low WSS, and these areas extended at follow-up. This trend was also seen in P2. Higher OSI (and lower WSS) were found below the impingement region, small bleb regions, and in other regions that experienced growth (Figure 7e,f). The average OSI values increased at follow-up by 31% per year (Table 2).

### 3.4. P4 Aneurysm

P4 grew in volume 170% per year, with noticeable growth in height and width (Figure 8). One daughter bleb may have formed across from the impingement region (Figure 8). However, this was the smallest initial aneurysm in this study, which may explain the large annual growth.

Initial time point pathlines suggest an elongated swirling flow through the aneurysm, while at follow-up, a larger rotational flow structure appeared (Figure 9a,b). KE increased by 131% per year between the two time points.

WSS contours in Figure 9c,d show elevated values at the neck of the aneurysm in the region of impingement at both time points. The black arrow in Figure 9d indicates the impingement region. The WSS gradient and the maximum value in this region both increased from the initial to the final time point. Large growth occurred in areas of low WSS (black circle in Figure 9d). Initially, an elevated OSI was only found near the impingement region (Figure 9e), but at follow-up, the OSI was elevated over a large area of the bleb as well as around the impingement region (Figure 9f).

## 4. Discussion

Previous studies found that large growth occurred in areas of low WSS and high OSI [9,10]. Our study also found that general aneurysm growth occurred in areas of low WSS. As the aneurysms grew, we noticed that the low WSS regions also grew. In addition, significant aneurysm growth occurred in areas of high velocity gradient between the wall and large swirling flow structures. Contrary to the previous studies, we found that high OSI areas did not necessarily associate to areas of general growth. Instead, large geometric growth often occurred in areas of low OSI.

Previous studies found that high WSS and variable gradients caused shape remodeling near the neck region impingement zones [2,10,11]. Our results showed that shape remodeling near neck impingement regions, unlike general geometry growth, occurred in areas of larger WSS, larger OSI, and large gradients of these variables.

Much of the literature has investigated aneurysm initiation and rupture, not growth, and used geometries only from one time point [2,4,7,12,13,14]. One study that examined aneurysm growth analyzed seven different patient geometries, at initial and follow-up time points, and found that aneurysm growth was associated with low WSS [9]. Our study confirmed this finding. Another study that used particle image velocimetry analyzed a cerebral aneurysm before and after one year of growth to find that there was no quantitative difference between the flow structures. The only major change was that the initial high WSS at the bleb decreased at follow-up [15]. Our study showed that changes in flow structures varied based on the amount of growth and the inflow type into the aneurysm. In P1–P3, we observed minor changes in the flow structures as aneurysms grew, but in P4, we saw significant flow structure changes during the growth period.

An additional observation that resulted from this study was a correlation between the change in aneurysm volume and the change in total KE for all four aneurysms (Table 2). In particular, the changes in volume and KE in aneurysms P2–P4 were very similar.

## 5. Conclusions

Substantial aneurysm growth occurred in areas of low WSS and high velocity gradients between the aneurysm wall and large swirling flow structures. Wall remodeling in the neck impingement region occurred near areas of high WSS, velocity, OSI, and large gradients of all these variables. All aneurysms showed an increase in KE from the first time point to the second, and this change in KE seems correlated to the change in aneurysm volume.

## Figures and Tables

**Figure 1 brainsci-11-00520-f001:**
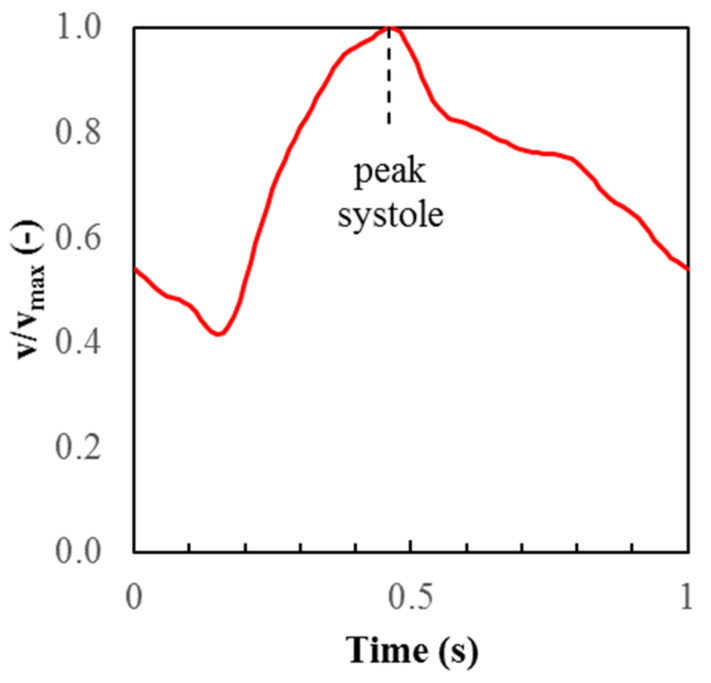
Normalized inlet centerline velocity profile throughout the cardiac cycle.

**Figure 2 brainsci-11-00520-f002:**
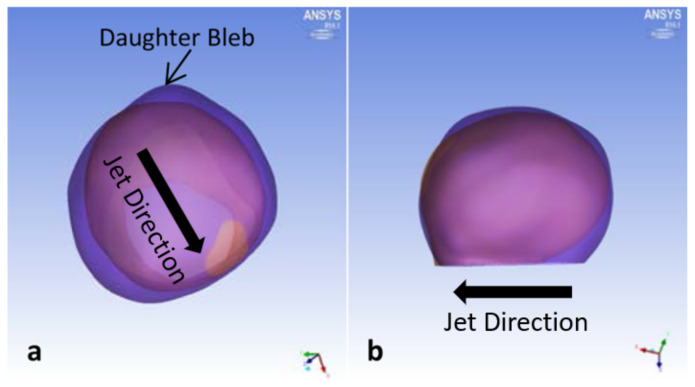
P1 aneurysm growth comparisons between initial diagnosis and follow-up times. (**a**) Top view and (**b**) side view, with block arrows indicating direction of impingement jet.

**Figure 3 brainsci-11-00520-f003:**
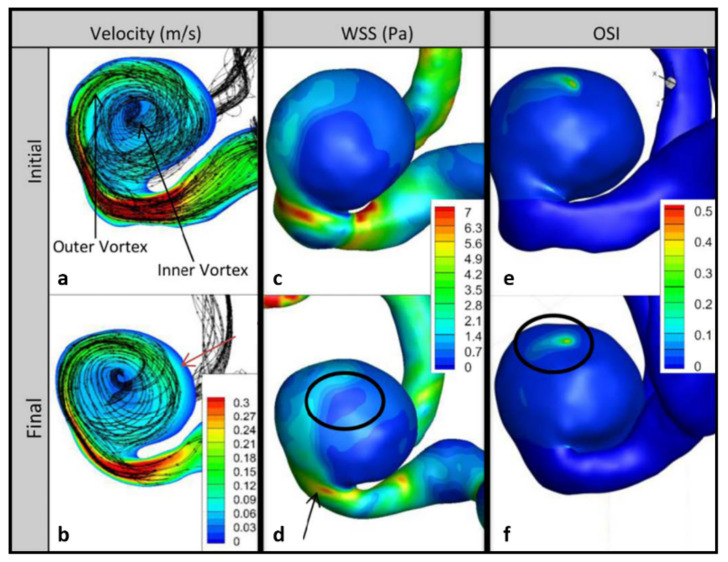
Impingement data planes and wall contours for P1. Velocity column shows changes in velocity between initial (**a**) and follow-up (**b**); red arrow shows region of high velocity gradient where growth occurred. Changes in wall shear stress (WSS) between initial (**c**) and follow-up (**d**); black arrow shows change in WSS near the impingement region and circle shows increase in WSS gradients near low flow. Changes in oscillatory shear index (OSI) between initial (**e**) and follow-up (**f**); circle shows decrease in area of elevated OSI near low flow.

**Figure 4 brainsci-11-00520-f004:**
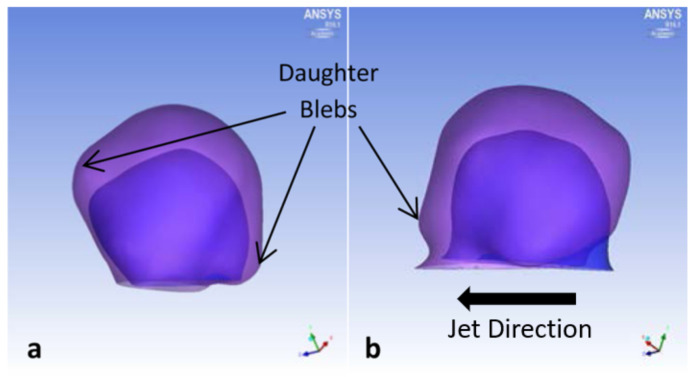
P2 aneurysm growth comparisons between initial diagnosis and follow-up times. (**a**) Side view plane normal to impingement jet, and (**b**) side view plane parallel to impingement jet. Block arrow indicates direction of impingement jet.

**Figure 5 brainsci-11-00520-f005:**
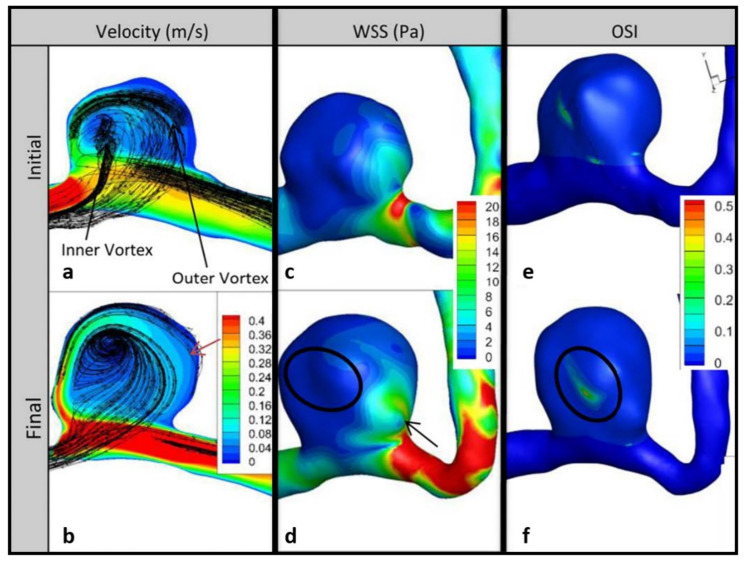
Impingement data planes and wall contours for P2. Velocity column shows changes in velocity between initial (**a**) and follow-up (**b**); red arrows show region of low velocity near higher velocity where growth occurred. WSS column shows changes in WSS between initial (**c**) and follow-up (**d**); black arrow shows change in WSS near the impingement region and circle shows low WSS near daughter bleb formation. OSI column shows changes in OSI between initial (**e**) and follow-up (**f**), circle shows increase in high OSI region.

**Figure 6 brainsci-11-00520-f006:**
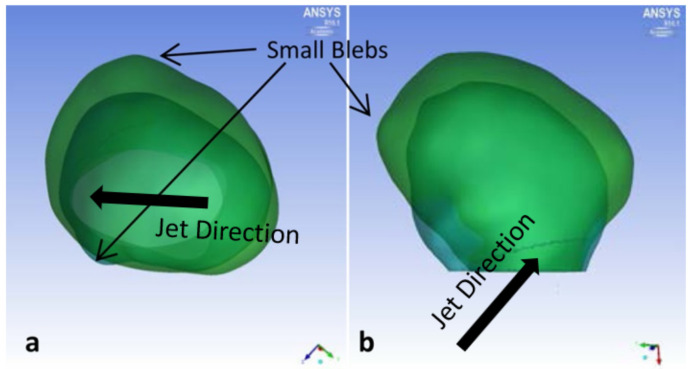
P3 aneurysm growth comparisons between initial diagnosis and follow-up times. (**a**) Top view and (**b**) side view, with block arrows indicating direction of impingement jet.

**Figure 7 brainsci-11-00520-f007:**
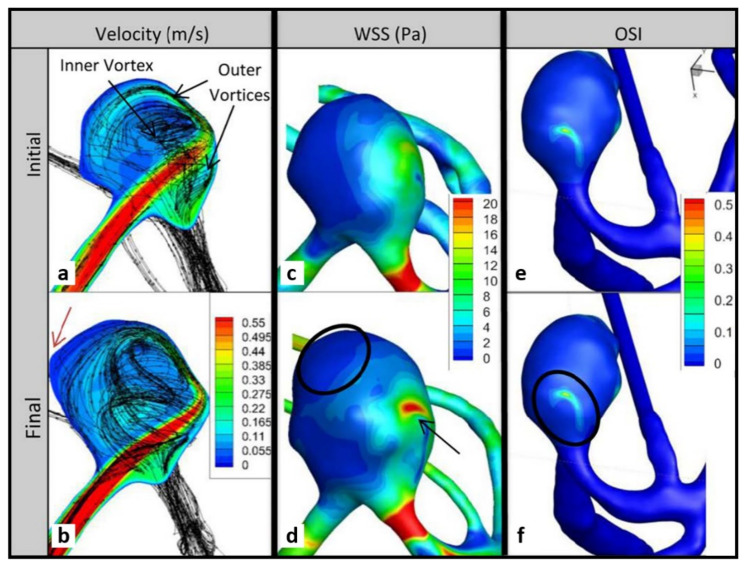
Impingement data planes and wall contours for P3. Velocity column shows changes in velocity between (**a**) initial and (**b**) follow-up; red arrow shows region of low velocity near higher velocity where growth occurred. WSS: changes in WSS between initial (**c**) and follow-up (**d**); black arrow shows change in WSS near the impingement region and circle shows region of significant growth. OSI: changes in OSI between initial (**e**) and follow-up (**f**).

**Figure 8 brainsci-11-00520-f008:**
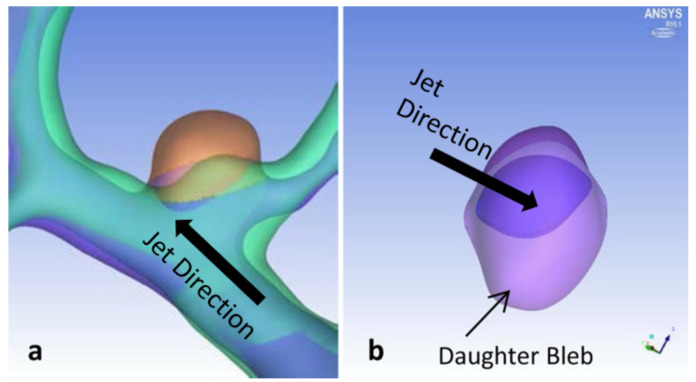
P4 aneurysm growth comparisons between initial diagnosis and follow-up times. (**a**) Top view and (**b**) side view, with block arrows indicating direction of impingement jet.

**Figure 9 brainsci-11-00520-f009:**
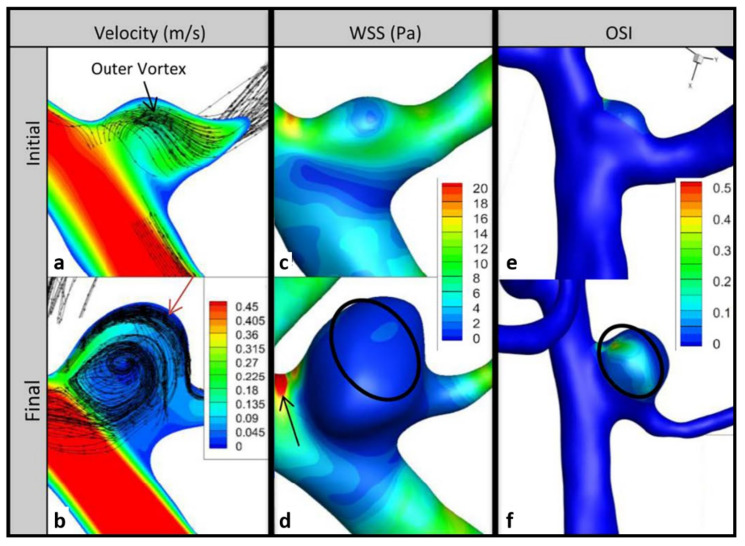
Impingement data planes and wall contours for P4. Velocity: changes in velocity between initial (**a**) and follow-up (**b**); red arrow shows region of low velocity near higher velocity where growth occurred. WSS: changes in WSS between initial (**c**) and follow-up (**d**); black arrow shows change in WSS near the impingement region and circle shows region with significant decrease in WSS. OSI: changes in OSI between initial (**e**) and follow-up (**f**); circle shows increase in area of elevated OSI.

**Table 1 brainsci-11-00520-t001:** Aneurysm geometry information.

ID	Location	Initial Size (mm)	Final Size (mm)	Initial MRA Year	Final MRA Year	Growth Time (Years)
**P1**	ICA dx.	8	9	2008	2010	1.08
**P2**	ACom and ICA sin.	5	7	2005	2011	5.96
**P3**	MCA dx.	8	9	2009	2011	2.64
**P4**	MCA dx.	0.5	2	2007	2015	7.33

**Table 2 brainsci-11-00520-t002:** Summary of changes in hemodynamic and morphological parameters in each aneurysm. Changes listed are average values per year of aneurysm growth (see Table 1).

ID	Hemodynamic Changes			Morphological Changes				
	WSS	OSI	KE	Volume	SR	AR	EI	NSI
**P1**	−27%	−33%	256%	61%	3%	−5%	36%	−35%
**P2**	0%	9%	56%	56%	2%	6%	23%	−12%
**P3**	−2%	31%	7%	11%	6%	0%	24%	6%
**P4**	−3%	6%	131%	170%	49%	15%	−11%	−10%

## Data Availability

The data used in this study were obtained from an internal Mayo Clinic database. This data is not publicly available.

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
