# Peer review of "Morphological and Hemodynamic Changes during Cerebral Aneurysm Growth"

_brainsci, 2021, doi:10.3390/brainsci11040520_

Round 1

Reviewer 1 Report

The study aims to analyze hemodynamic conditions inside IA and correlate changes with aneurysm growth using  patient-specific CFD simulations and a patient longitudinal cohort of 4 patients.  Although the mythology is relatively new, it is not a novel approach in the study of intracranial aneurysms. Also, qualitative comparison between morphological and hemodynamic parameters; which has been extensively done in this type of analysis, lower significantly the interest in this publication.

In addition,  manuscript quality needs to be improved significantly.

concerns:

  • Introduction: The study needs to be better justified. It is not clear to me the importance and novelty of the study.
  • Material and methods:

  1. Lack of patient specific or more realistic boundary conditions
  2. The paper should define each hemodynamic metric to be used
  3. Morphological parameters have not been defined.

  • Results:

1.Pressure shown in figures does not seems to be correct ( 525 Pa = 4 mmHg).

  1. Figures does not explain whether this images are averaged of taken at either systolic or diastolic time points.
  2. Volumetric flow rates used in the simulations need to be reported.
  3. To the best of the author’s knowledge, vortex cannot be visualized using streakline.
  4. OSI ranges between 0 and 0.5. Therefore, figures are misleading.

Author Response

Please find the response attached. Our responses to these comments are shown in blue.

Reviewer 2 Report

This is an interesting computational study characterizing the morphological and hemodynamic changes during cerebral aneurysm growth. The authors assessed the changes in four different patients with different size (or parameters) of aneurysm growth. As described in details in the Results section, the characteristics of each shape remodeling vary depending on the size and years of growth. My only comment is to include a table summarizing the properties of each growth in P1-P4, allowing readers to have a general idea of the obvious characteristics of each respective growth. 

Author Response

(The authors gave the same response as above.)

Round 2

Reviewer 1 Report

Page 5:  Please include the volumetric waveform used as inlet boundary.

Page 8: Pathlines showed a larger vortex “. To the best of the author's understanding, vortices cannot be educted using pathlines. Please rephrase. 

Page 8: “ Pathlines showed a larger vortex “. To the best of the authors understanding, vortices cannot be educted using pathlines. Pleas rephrase.

Page 8 “An area of low pressure and WSS was found on the top side of the aneurysmis seen near the center of the flow vortex”.  Vortex cannot be visualized using streamlines, please rephrase.

Page 12: “Vortices”.

Author Response

Please find the response attached.
